# New Micromobility Means of Transport: An Analysis of E-Scooter Users’ Behaviour in Trondheim

**DOI:** 10.3390/ijerph19127374

**Published:** 2022-06-16

**Authors:** Margherita Pazzini, Leonardo Cameli, Claudio Lantieri, Valeria Vignali, Giulio Dondi, Thomas Jonsson

**Affiliations:** 1Department of Civil, Chemical, Environmental and Materials Engineering (DICAM), University of Bologna, Viale Risorgimento, 2, 40136 Bologna, Italy; leonardo.cameli2@unibo.it (L.C.); claudio.lantieri2@unibo.it (C.L.); valeria.vignali@unibo.it (V.V.); giulio.dondi@unibo.it (G.D.); 2Department of Civil and Environmental Engineering, Norwegian University of Science and Technology (NTNU), Høgskoleringen 7A, Gløshaugen, NO-7491 Trondheim, Norway

**Keywords:** e-scooter, vulnerable road users behaviour, speed analysis, path choice

## Abstract

Negative effects of a massive use of cars, such as congestion, air pollution, noise, and traffic injuries, are affecting the cities everywhere. Recently introduced shared vehicles, such as e-scooters and electric bicycles, could potentially accelerate the transition towards sustainable mobility. Although these vehicles are becoming increasingly common and accepted within regulatory frameworks, some local governments are not yet ready to integrate e-scooters into their transport systems. Indeed, the legislation is unclear as it is not easy to determine whether the e-scooter is more like a bicycle or a vehicle. Moreover, it is difficult to predict the impact of e-scooters on road traffic, as well as the type of road infrastructure chosen by e-scooter drivers or the possible interaction of such vehicles with weak road users, such as pedestrians or cyclists. This study showed an analysis of speed and behaviour of e-scooter drivers in the city of Trondheim (Norway) to investigate how to manage this mode of transport. A total of 204 e-scooters were observed on six different roads in the city centre. The speed of e-scooter drivers was measured by a speed tracker (average value 15.4 km/h) and their behaviour recorded by a hidden observer in the field. Gender, age, distance from pedestrians, speed adaptation to the environment, and type of vehicle used were registered for each e-scooter. Through a Binomial Logit analysis, the data obtained were used to analyse the type of road infrastructure preferred by e-scooter drivers. Results showed that the cycle path is more widely used with percentage value from 60% to 90% of users. In addition, the probability of choice depended mainly on the road environment. The aim of this analysis was to assist local authorities in regulating the safe use of e-scooters and developing appropriate policies for their integration into cities.

## 1. Introduction

In urban communities the popularity and the use of micromobility is rapidly increasing, especially for the “last-mile” transportation solutions. Last mile transportation covers the gap between conventional transportation hubs (train station, bus stops, modal interchange parking spots, etc.) and the final destinations, especially in cases where infrastructures are very crowded [1]. In the report written by the International Transport Forum (ITF), micromobility is defined as: “[…] the use of micro-vehicles: vehicles with a mass of no more than 350 kg (771 lb) and a design speed no higher than 45 km/h”. This definition includes both human-powered and electrically assisted vehicles such as bicycles, e-bikes, kick scooters, and e-scooters but also skateboards, one-wheeled balancing boards, and four-wheeled electric micro-vehicles [2]. Micromobility can offer flexibility and efficient door-to-door accessibility, while public transport is characterised by higher speeds and greater spatial reach. The resulting synergy of high speed (and thus spatial reach) of public transport with the door-to-door accessibility provided by micromobility creates a degree of access, speed, and comfort that can compete with that of private motorised vehicles [3]. Therefore, this combination makes modal shifts more attractive as well as potentially contributing to making cities more liveable, less congested and with reduced levels of air and noise pollution. In order to fully exploit the potential of micromobility towards a transition to more sustainable urban mobility systems, it is essential to analyse and study micromobility in the context of access to the first and last mile to and from public transport [4]. The concept of micromobility is constantly evolving. For this reason, the definition of micromobility is rather broad, as it is designed to be tested in the future and cannot be limited to a certain type of vehicle or source of energy. It can be used to facilitate the regulation of new vehicles placed on the market and to create a category that includes all micro-vehicles regardless of the vehicle characteristics, such as number of wheels or driving position [2]. The “E-scooter” is classified as a type of vehicle called powered standing scooter. This one must satisfy some conditions to enter in the micromobility category: weight below 500 kg, provision of a motor, and availability as a shared service. The speed range of the e-scooter, which makes its driving force, is strictly dependent on the battery of the model, but usually it is between 12–40 km/h [5]. This vehicle is smaller (occupying less space on the streets), lighter (making its transport easier) than a bicycle, and requires less maintenance. However, e-scooters have small batteries (i.e., less travel autonomy) and if they run out, there are no alternative ways to use them. The lack of pedals reduces flexibility and makes them a less healthy transport mode for users [6]. The two main types of shared systems are station-based and dockless sharing systems. While in a station-based system users can start or end their trips only at predefined locations, the dockless system allows users to start or end their trips (almost) anywhere in the city. Actually, e-scooter regulations differ widely among jurisdictions, with effects on the extent and nature of safety issues. It is important to establish the relationship between dockless e-scooters and existing bike sharing in order to be able to decide on fair transport policies and the infrastructures to be created including both bikes and e-scooters. For example, in Washington D.C., the spatial and temporal patterns of dockless e-scooters were explored to define trip origins and destinations. In addition, these data were compared to those of traditional docked bike-sharing services. Bikes are primarily used by individuals commuting to and from work, while e-scooters are used for leisure and tourist trips [7]. As for the type of use of an e-scooter based on its purpose, research in Austin (TX) showed that users tend to ride both bicycles and e-scooters with a lower average speed for leisure purposes (i.e., over weekends and off-hours) than during commuting or for other transport purposes (e.g., on weekdays and during working hours). The e-scooter speed, especially for transport purposes, is as high as the speed of a bike, but the results show that the average travel speed for bicycles (12 km/h) is slightly higher than the average travel speed for e-scooters (10 km/h) [8]. To prove the link of these two modes of transport, a study carried out in Chicago analysed the data of two sharing companies. The results proved that users of bike sharing decreased by 10.2% in the area reserved for shared vehicles with the introduction of shared e-scooters [9]. However, the real difference between these two transport modes lies in safety. In fact, focusing on the accidents, a study in Salt Lake City showed a significant increase since the arrival of e-scooter in the urban context. The largest number of types of injuries comprised orthopaedic and cranial injuries. With an increasing number of e-scooters and the spread of their use, these injuries are expected to increase as well. [10,11]. Another study confirmed that this increase in injuries results from the illegal and risky behaviour of both shared and private e-scooters [12]. Considering the danger associated with the use of e-scooters, Europe is trying to find rules for e-scooters, but unfortunately there is no ‘conformity’ or union within the countries of the European Union. A report of the European situation [13] was realised by the Forum of European Road Safety Research Institutes (FERSI), a non-profit organisation forming a flexible network of European road safety research organisations. A short questionnaire was created to make an inventory of the information available in different countries, in terms of legal status, use, and safety. Norway, Italy, and Denmark classify e-scooters as bicycles. In Switzerland, Portugal, and Sweden the legal e-scooter category depends on the maximum speed of the e-scooter. In Austria, Belgium, Germany, France, and Spain e-scooters are considered as a separate category regulated by special laws, but, in general, the rules for bicycles have been applied for them. A particular situation is in Belgium where e-scooters belong to the category of ‘personal transportation devices’ with two sub-categories, motorised and non-motorised. These devices cannot be wider than 1 m and motorised devices, which e-scooters belong to, cannot go faster than 25 km/h. In all the countries considered, except for Hungary, there is a general maximum speed limit for electric scooters of 20 or 25 km/h. In Italy, the maximum speed limit is 20 km/h on mixed paths, both pedestrian and cycling, and 6 km/h in pedestrian areas. In Finland and the Czech Republic, the maximum speed limit is 25 km/h. If the e-scooter can travel at higher speeds, it is classified as a moped. In addition, in France, the speed of the e-scooter is limited to 25 km/h, but its speed limit is 45 km/h on the roads. In Sweden, there are three categories for e-scooters based on their maximum speed. If the e-scooter can reach 20 km/h, it is classified as a bicycle, 25 km/h it is classified as a class I moped, and 45 km/h classified as a class II moped. This survey highlights how countries are working to find a legal status for this means of transport and are still working to develop more targeted or specific legislation.

Authorities have to face important problems related to the use of e-scooters in the city, namely safety problems and urban space use issues. Moving on the empiric point of view, this research aimed to contribute to the literature through data obtained from field analyses of speed and behaviour, in terms of choice and interaction with the other weak users of the road. Moreover, the identification of the factors choice of the path may help local politicians to develop policies and rules based on concrete evidence and mobility planners to develop actions for better integration of e-scooters in cities.

## 2. Materials and Methods

Norway has been a e-scooter development hub, especially in Oslo where 17,984 e-scooters cover a service area equal to 246.09 km^2^ (the second highest in Europe) [14]. Taking this into account, Trondheim is the third largest city in Norway by number of inhabitants with a high percentage of students, the right place to experiment on the use of shared e-scooters by a private company, the Ryde Technology AS. In summer 2019, this company distributed about 200 e-scooters around the city [15]. The measurements analysed in this study were taken within one month, from mid-September to mid-October 2019, between 8:00 a.m. and 6:00 p.m. with the peak time between 1:00 and 2:00 p.m. Six different road sections were defined in the city center of Trondheim. The geometric and functional characteristics of each road were analysed through an instrument consisting of a wheel with a counter measuring the distance travelled according to the number of revolutions completed by the wheel. The different roads are the following ones (Figure 1):Elgeseter, with a cycle path and a sidewalk on each side. This is an important bridge connecting the city center and the other part of the city;Olav Tryggvasons Gate, also with a cycle path and sidewalk on each side. This carriageway has a lane for each direction on which buses, trucks, and cars pass;Nordre Gate, a pedestrian street with a lot of interactions between pedestrians and cyclists; it is crowded in the afternoon when people leave work and schools;Innherredsveien, an important connection between the suburbs and the center, consisting of a sidewalk and a cycle path;Øvre Alle, a one-lane road with a sidewalk on one side; it connects NTNU University with some schools in the neighbourhood;Munkegata street, with two-lanes, parking spots, and two wide sidewalks for each side.

A traffic radar (Genesis VP Directional) was used to measure the speed of e-scooters. The radar instrument was installed in a hidden point, adjacent to the road section so as not to affect the user’s behaviour. For precise measurements, the angle of inclination of the radar, regarding the direction of the users’ motion, was as small as possible. If necessary, the measured speed based on the angle was corrected according to the manual instrument (Figure 2).

During the e-scooter ride, the observer collected other information to define the type of user and their behavior such as:Gender and age of driver divided into two groups, 18–35 and >35 years.Type of e-scooter used (shared or private).The path used by e-scooter riders (cycle path, cycle lane, sidewalk, pedestrian zone, or roadway).Number of people on the e-scooter at the same moment.The distance observed between the e-scooter and the other road users when overtaking. Overtaking is defined as the process of overcoming a slower means of transport or a person travelling in the same direction. Distances were measured at the beginning of the overtaking. The observer fixed some reference points on the analysed section, empirically defining the distances and measuring them. The differences were divided into four categories: no interaction with others, distance 1 m, distance 50 cm, and distance 30 cm (Figure 3).Driver behaviour in the presence of other users. It was divided into three categories: straight, when the e-scooter’s users go straight without any influence from obstacles although these are very close; “zig-zag”, when the drivers do not decrease the speed to avoid or to overcome obstacles. The last category includes drivers who reduce speed in the presence of other users (Figure 4).


### 2.1. Statistical Analysis of the Data

#### 2.1.1. Generalised Linear Model and Student *t*-Test

The Generalised Linear Model (GLM) was used to establish the relationship between the factors defined above (user characteristics, crowding, etc.) and speed (dependent variable) with the SPSS program. Thanks to GLM, the dependent variable was linearly related to the factors via a specified link function that described the relationship between the linear predictor and the mean of the distribution function. This ordinary linear regression predicted the expected value of a given unknown quantity (the dependent variable) as a linear combination of a set of observed values (predictors). When the dependent variable had a normal distribution, this model was appropriate to describe a random variable with real values that tended to concentrate around a single mean value. To estimate the parameters, the GLM used the Maximum likelihood estimation: this method estimated the parameters to maximise the probability of obtaining values of the depending variable on the values of the independent ones. Initially, after uploading the observed data and defining the variable types, the linear scale response could be chosen in SPSS, specifying Normal distribution and Identity as a connection function (*Xβ* = *μ*; where *X* is the independent variable, *β* is the linear combination of unknown parameters, and *μ* is the average distribution). After that, the program wanted to know which variable was the dependent one (speed). The aim was to understand how and how much the speed changed in accordance with the other variables present in the model. Type I and Type III analysis were chosen because the former was generally appropriate when there was a fixed reason for ordering predictors in the model, while type III was more generally applicable. After these steps, the program was ready to calculate the results. For Type I, the 95% confidence interval was calculated for one factor at a time. The width of this range was directly related to the number of available data. According to previous research found in literature [16], through the Student’s *t*-Test it was possible to perform a test of statistical hypotheses in order to analyse any correlations between different categories within the same variable. The test was used to determinate whether the average values and the standard deviation of two datasets were significantly different from each other. The statistic t was calculated as follows:(1)$$t=\frac{B_{1}-B_{2}}{\sqrt{se_{1}^{2}-se_{2}^{2}}}=\{\begin{array}{c} <1.96\;no\;significant-present\;some\;correlations\\ >1.96\;significant-the\;means\;are\;different \end{array}$$
*B*_1_ and *B*_2_ are values returned by the model for each variable.*Se*_1_ e *Se*_2_: standard error returned by the model for each category.

It is important to assess that the probability of rejecting the null hypothesis when it was true (the level of significance) for each category was equal to 0.001. The significance level (sig.) of 0.001 indicated a 0.001% risk to detect a difference when there was no actual difference. This was because speed had to be higher than 0 km/h. In fact, lower levels of significance indicated that stronger evidence was required before rejecting the null hypothesis. In addition, this test allowed to find statistical differences or similarities within the same factor between the different categories considered. By analysing the correlation of the variables, it was possible to determine three contexts:A complete overlap when the two categories were statistically similar;A complete separation when the two categories were statistically different;A partial overlap when the two analysed categories could be statistically similar or different. In this case, it was necessary to study carefully if there was a correlation.

#### 2.1.2. Statistical Analysis of the Collected Data

The statistical analysis of the collected data showed the relation between categories inside the same variable. The aim was to understand how these categories were related and if the separation from the reference level could be relevant. The reference level was set on a young female rider who travelled on a private scooter without interactions with other road users in a pedestrian zone with also a cycle path present in Elgeseter Blu at a speed of 13.96 km/h. In this analysis, the SPSS program had information on the dependent variable (speed), categorical variable information, and the characteristics of the model as input. In output the model defined
Test of model effect, (Table 1) where the level of significance (sig.) of each factor considered in the model was defined for analysis type I and III.Parameter estimates, a table (Table 2) providing for each parameter the intercept (reference level speed), the speed value in relation to the reference category, the 95% confidence interval, and the hypothesis test with the significance value. The last element showed how much a category within the variable was statistically significant in relation to the reference category.

#### 2.1.3. Backward Analysis

With SPSS it was also possible to carry out a backward analysis. This is a technique to analyse randomised algorithms from output-to-input. This process started with the statistical analysis of collected data (parameter estimates, Table 2) with all the internal variables. At each step, the GLM ran without the variable with the highest sig. value, which was statistically insignificant for the analysis. Table 3 shows an example of the test of model effect obtained after each step. The intercept showed the speed of the reference variable, and the B value was returned by the model for each variable.

It was possible to know what the most significant variables for the model were. After removing these variables, a change might occur in B value of the intercept or of other variables. Variables with the sig. value less than 0.06 were statistically significant while in other cases variables were not statistically significant. Although the check value was usually 0.05, 0.06 was chosen because the amount of data was not so wide, and the significance level had to be determined based on the context of the experiment performed. The value of 0.06 indicated that type I error was likely to be committed with 6% probability. This type of error occurred when a wrong hypothesis was made, and it was accepted [17].

#### 2.1.4. Binomial Logit Model for the Pavement Choice

The binomial logit model was used to calculate the choice probability of using the sidewalk, a cycle path, or lane or the roadway by the e-scooter users. The purpose of this analysis was to understand how the choices of e-scooters were distributed. The binomial logit analysis was focused on the roads with sidewalks and cycle paths. Only these two kinds of infrastructures were chosen in order to analyse the interactions with the other road’s users such as pedestrians and cyclists (only the data related to Elgeseter Bru, Olav Tryggvasons Gata, and Inherredsveien were considered). The logit analysis was possible with the SPSS software choosing the path as a dependent variable. The binomial logit method calculated the probability for an observation to fall into one or two categories of a dependent variable (in our case the probability to choose the cycle path or the sidewalk) based on one or more independent variables that could be continuous or relevant to the category (Street, Gender, Age and AddInfo). Therefore, the dependent variable was the path, the probability distribution chosen was the bi-nomial one, and the link function was the logit one. With the data and information on the categories obtained, it was possible to carry out tests of different possible effects on the model. These tests showed the significance level of each factor considered for the model, the probability of choosing the type of infrastructure, the standard error, and the significance value for each category. The software took the sidewalk as a reference level and the probability calculated was referred to the choice to ride on the sidewalk. The estimation of the parameters was followed by the estimation of the probability. The following formula of the Logit model should be used:(2)$$P=\frac{e^{\beta x}}{1+e^{\beta x}}$$
where *P* is probability to choose the infrastructure and *βx* is given by the sum of all the parameters returned by the program and included in the choice of that infrastructure with those specific conditions $\left(\beta x=\overset{n}{\underset{i}{\sum }}\beta_{i}\right)$. A backward analysis highlighted which variables influenced the choice of the e-scooter users.

In conclusion, the Figure 5 shows the flow chart about the used methodology implemented in this research.

## 3. Results

### 3.1. E-Scooter Users

A total of 204 e-scooters were observed (Table 4) and the percentage of the different categories of e-scooter users was collected.

The table shows that in Trondheim most e-scooter users are men, although women were present (31.9%) too. As for the age, e-scooters are mostly used by young people. Comparing the results of this research with a Parisian survey, the percentage of men and women was the same (69% of men and 31% of women) as well as for the age (86% were under 35 years old and 14% other ones) [18]. This survey confirmed the trend of Trondheim, but the number of women was higher than in the Parisian context. Therefore, the most common e-scooter user is a man aged between 18 and 35 years old with a shared e-scooter, travelling without interaction with other road users and proceeding straight while moving on the cycle path.

### 3.2. Speed Analysis from Observed Data

By focussing on speed, it was possible to determine the average speed and the Standard Deviation showed no large variations of the speed value (Table 5). In addition, the speed distribution and normal distribution associated from the speed dataset recorded on the field were analyzed.

By comparing speed with the age and gender of users, it was possible to see that male users aged 18–35 years represented the fastest category. This result is in accordance with a study in China, focused on the red-light running behaviour. It showed that age and gender were significant variables to predict the behaviour of drivers at traffic lights with red lights. In fact, young men aged 18–35 and middle-aged men drivers were more likely to accelerate than older people in front of the red light [19]. In addition, users who proceeded straight maintained a higher speed than the other categories (15.80 km/h), followed by the zigzag (14.58 km/h) and speed-reducing users (12.82 km/h) when meeting other road users. With reference to the chosen path, the highest speed was recorded on the road (17.21 km/h) followed by the cycle path (16.59 km/h). Users on the pedestrian zone or sidewalk were the slowest (13.49 km/h and 14.61 km/h respectively). This is evidenced by the gap between e-scooter and pedestrian speed. In fact, a study on the speed pedestrian at the crossing [20] showed that the users speed fluctuated from 2 to 8.5 km for hours. So, the gap between the two categories was equal to 10 km/h. The behaviour of e-scooters drivers in vicinity of a pedestrian route was analysed in relation to the measured speed, comparing the data on the crowding and different paths (Table 6). As for the path, the pedestrian zone, the cycle path and sidewalk were the only ones to vary because they were the only ones with interactions. Table 6 shows the relations between the above categories and the speed:on the cycle path, the users with a distance lower than 50 cm from the other road users during the overtaking were the fastest (16 km/h);in the pedestrian zone the rider driving 1 m from another road user, during overtaking, was the fastest. The other two categories held the same average speed. One explanation might be that the driver overtaking another road user at a distance of less than 50 cm did so during rush hours, when the road was busier, and people were driving faster as they were in a hurry;on the sidewalk the difference between the average speeds of the two different categories, without interaction and with 50 cm, was equal to 2 km/h. One explanation might be that the driver in a crowded situation usually reduced the speed to avoid interactions and to overtake others with a shorter distance than in other cases. Those who drove keeping 1 m from other road users were as fast as those who drove without finding interactions. In fact, the first category was made up of drivers who were in a hurry and did not reduce the speed but overtook the others maintaining a good distance.

### 3.3. Speed Statistical Analysis from Observed Data

Trough GLM analysis it was possible to calculate the 95% confidence interval and the statistical hypothesis with the Student’s *t*-Test for each variable of categories (Table 7). This statistical analysis underlined the possible relation between the independent variables and the dependent one (speed). Table 8 shows the average speed, the standard error, the 95% confidence interval, and the value of significance for each path.

Focusing on the relation between the different paths detected by Student’s *t*-test (Table 8), some categories revealed statistically more significant differences compared with others:Sidewalk–Roadway: these two categories were statistically different with a *t*-Test value higher than 1.96. This was because those who drove on the road had to pay attention to many things compared with those who proceeded on the sidewalk. The same can be said for the relationship between sidewalk and cycle path;Roadway–Pedestrian zone: these two intervals were separated because the speed and the behaviour of riders were statistically different;Pedestrian zone–Cycle path: these two intervals were separated. Differences could be found in the different environments and related behaviours this type of path entails as the bike path is included in the road.

The next step was the Statistical Analysis of the collected data used to understand how the speed and the other variables analysed were related and whether the difference with the reference level could be relevant (Figure 6). Analysing the most important variables, the following considerations can be made:Street: Inherredsveien was the fastest street because the section chosen for the analysis was on a small hill and people could run faster than in the plain. The slowest road was Nordre Gata because this was a pedestrian zone and speed had to be reduced to avoid interaction with other road users;Crowding: the relationships between the categories did not change too much. The fastest category was those of riders overtaking other road users with a certain distance (1 m) followed by users without interactions. The slowest consisted of those who overtook with less than 50 cm from the others.Behaviour: the difference between the fastest and the slowest speed was very relevant in this case, 2 km/h. The same trend was shown by the simple analysis of the data considering one factor at a time;Path: the user on the road was faster than on the other paths but the slowest was the user who rode on a sidewalk due to interactions with pedestrians.

A backward analysis was carried out to check the influence of the variables considered (such as path, gender etc.) on the speed.

The variable with the highest significant value (sig.) was excluded at each step. When the significant value of the variable was 0.00 it meant that the variable was statistically significant for the analysis. The removal of the variable “Street” implied a reduction of the significant value for Behaviour, Path, and AddInfo. On the other hand, the removal of Gender, Age, and Crowding led to an increase of the significance value. There were changes of speeds for the Path categories: those who rode on the roadway were the fastest followed by the cycle path users and sidewalk users. People in the pedestrian zone were the slowest. After that, the categories “Additional Information”, “Age”, Number of people, and Gender were removed. This did not involve a large variation of the significance value of the different variables and the changes of the other variables were very limited since these factors were not so relevant in the model. The next variable to be erased from the analysis was “Behaviour”. In this case, the crowding was the only variable that improved its significance value. The only thing changing was the order of different speeds within the crowding factor: those who overcame with distance 1 m were the fastest followed by the users with no interactions and the users with 50 cm distance when overtaking. The statistically significant variables for the model were Crowding and Path (Table 9) with a low significance value.

### 3.4. Analysis of the Choice Infrastructure Percentage by Users

It was then possible to calculate the probability to choose the sidewalk or the cycle path considering the variables Street, Gender, Age, and AddInfo referred to the analysed section. The result of the binomial logit model for the three streets with these different infrastructures (131 measurements) were as follows:For the two streets with a good separation between the roadway and the cycle path (Inherredsveien and Elgeseter) all e-scooter users preferred to choose the cycle path (Table 10). This was also supported by the backward analysis because the program returned the same trend regardless of the variables considered within the model (Table 10, last column).For the street with a cycle lane adjacent to the roadway without any physical separation, it was possible to see a different behaviour with reference to the age of the users: a higher percentage of young people (18–35 years) chose to ride on the sidewalk (Table 11); the opposite was for the category > 35 years old. A backward analysis showed that for this street the only variable involving some changes was the Age. When street was the only variable, the results showed a higher probability to choose the sidewalk instead of the cycle lane (Table 11, last column).

A backward analysis was also carried out to find out which variables most influenced the choice of the type of infrastructure. Considering all the factors, the variable with the highest significant value was excluded at each step as statistically not significant for the analysis. In Inherredsveien and Elgeseter, not all categories created to differentiate the type of users were necessary. Depending on the model used, the users chose the type of infrastructure in relation to the geometry and the environment of the road considered. There were no differences between the different categories except for the value of the choice probability. The information about the different means of transport, shared or private, did not affect the choice of the favourite infrastructure for that model. The different types of e-scooter used had very little influence on the infrastructure chosen. After the backward analysis, it was possible to notice that the removal of the variables from the analysis did not involve major changes, especially for the significance value.

## 4. Discussion

E-scooter users analysed in Trondheim showed similarities within the European contest. In FERSI survey [13], the main users of different European countries were analysed. Half of the countries (such as Norway and Germany) indicated that young adults were the main users of e-scooters. The other half (including Italy and Spain) did not have information about what age group the users mainly belong to. Outside Europe, a survey in Minneapolis [21] showed the ratio between the percentage of the young population and the male and female and results were like those in Trondheim, namely that most e-scooter users were young men. This trend was confirmed by a questionnaire in Greece where the results showed that e-scooters increase the gender mobility gap instead of bridging it [22]. On the contrary, gender research in Seoul about the use of e-scooters showed greater uniformity with the percentage between men (52.6%) and women (47.4%) [23]. As for speed, the results obtained in this study on e-riders were compared with the average speeds of cyclists. Research in Sweden found that the average speed held by cyclists was between 12.5 during daytime and 26.5 km/h [24]. A similar result was measured in Italy [25] where it was shown that the average speeds of cyclists varied between 14.6 and 22 km/h. This result was also confirmed by research in San Jose, California [26], where e-scooter speed ranged between 14 and 18 km/h. So, the average speed of e-scooters (15.4 km/h) was about the same as that of cyclists in all the research. On the other side, a research study on the vibration given by the path showed that e-Scooter riders experienced more severe vibration impact than cyclists if they were running on the same facilities. So, the e-scooter riders were subject to increased safety challenges due to the increased vibrations, speed variations, and constrained riding environments [27]. It was also interesting to identify the characteristics of the road chosen and how it influenced the behaviour of drivers. The analysis showed that that Øvre Alle was the fastest street, probably because it was a small street connecting NTNU University with some schools in the neighbourhood, so it was particularly crowded only in the morning during peak hours (around 8:00 am). Nordre Gata was the slowest one maybe because it was a pedestrian zone shared by cyclists and pedestrians. In fact, e-scooter-users reduced speed in the presence of vulnerable users on the same way or in crowded conditions since they had to pay more attention to avoid interactions. Drivers on roadways were the fastest, followed by those who rode on the cycle path. Moreover, speed of e-scooters decreased during overtaking when the distance between them and other users decreased and when they found other road users on their way. This trend was confirmed by the behaviour of cyclists whose speeds were influenced by the presence of other road users. In fact, when the path was shared with pedestrians, the cyclists’ speed decreased accordingly [25,28]. The cyclist adjusted the speed when meeting pedestrians just like e-scooter drivers did [13]. Gender and Age, on the contrary, did not affect the speed of e-scooters. Statistical analysis carried out by SPSS software showed different results from the analysis of the measured data, possibly because the program considered all factors together, instead of each separate variable as the present study did with the data collected. On the contrary, as for the road infrastructure chosen, the characteristics of users, and the crowding of the road with respect to the speed, the same conclusions were reached as the analysis of the collected data. At the end of the backward analysis carried out thanks to GLM results, it was possible to conclude that the factors that mainly influenced the GLM model (and the speed accordingly) were Street and Behaviour. In fact, their removal led to big changes in the results of the analysis. Significant factors for the model were also Path and Crowding, fundamental to give relevance to the model when analysing the variation of speed. Finally, there were no relevant differences between the average speeds for the variables Gender, Age, and AddInfo. Analysing the type of road chosen, according to FERSI survey [13] about e-scooter riders in Europe, most countries replied that e-scooter drivers are likely to use bicycle facilities, if available. When not available, they are expected to use the road in Austria, France, Germany, Portugal, Sweden, and Switzerland if the speed limit on that road is not higher than 50 km/h. In Norway, e-scooters are allowed to use all parts of the road, and they are supposed to use the sidewalk only when pedestrian traffic is low and they are not dangerous for pedestrians. This research in Trondheim confirmed the European trend in the use of the cycle path. As in Inherredsveien and Elgeseter, the choice of the infrastructure was closely linked to the road environment and did not depend on other variables. The same behaviour was recorded in a study carried out in Virginia using the data collected by the Global Positioning System. The model results suggested e-scooter riders were willing to travel longer distances riding on cycle paths (59%), multi-use paths (28%), and the remaining percentage on other way [29]. Different results were recorded during the pandemic: e-scooter users were attracted to sidewalk infrastructure, even though curb use policies often prohibit it. This could be explained by the feeling of protection from the sidewalk given by the roadway [30]. The users’ speeds showed that in the section with pedestrian zone and cycle path or in the local roads the average speed was higher than that of a two-lane road (two lanes in two directions) with pavements, one on each side, as in Munkegata. The Age variable was fundamental for Olav Tryggvason Gata. In fact, according to the model, the over-35-year-old category preferred to use the cycle path while younger riders preferred the sidewalk, possibly because in this street, the cycle path was at the same level as the road without any protection and the sidewalk was very wide. Younger users chose to ride on the sidewalk at a lower speed than on the cycle path. As for the distance that e-scooters kept from the other road users during overtaking, and those who overtook with 1 m distance and those who had no interactions were the fastest.

## 5. Conclusions

Few data are available on e-scooter riders based on speed, their behaviour on the road, and safety. The concern of policy makers is increasing accordingly because of the lack of appropriate regulations for e-scooters. Although not such a large sample was analysed in Trondheim, from the analysis of data this study provided some answers that could help to adapt e-scooters into a regulatory framework with appropriate laws:Most e-scooter users are young men, but women are present too and the percentage is increasing.Gender and Age do not affect the speed of e-scooters.In terms of speed, there is not a great difference between bicycles (15–18 km/h) and e-scooters (15.4 km/h).Both cyclists and e-scooter drivers choose the infrastructure that makes them feel safer when travelling with limited interaction with weaker users.E-scooter riders prefer infrastructures separated from cars, such as cycle paths (90% and 60% in Inherredsveien and Elgeseter, respectively), bike lanes, or sidewalks (55% in Olav Tryggvasons Gate) which allow higher speed and less interactions.E-scooter users reduce speed in the presence of vulnerable users on the same way or in crowded conditions.The different types of e-scooters, shared or private, do not influence the choice of the infrastructure.

As said before, probably the number of analysed e-scooters, divided between shared and non-shared, could be increased in order to reach a greater precision in the analysis. Additional data on the relationship between the behaviour of e-scooters and interactions with other users and cyclists could be collected with a survey through the app used to rent e-scooters and installing cameras. Information about the busiest hours of the day, the average speed and the length of each trip may also be collected. In addition, it could be very interesting to simulate e-scooters in city-centre traffic through a microsimulation, especially in situations of extreme crowding. These data could be useful to deepen the problem of the distance maintained by e-scooter riders especially during overtaking. But they could also be useful to decide what solutions to take to improve the coexistence of e-scooters with pedestrians and cyclists, as well as the best places to install parking lots. Finally, e-scooters could replace non-active means of transport or very crowded public transport without major investments in public mobility.

## Figures and Tables

**Figure 1 ijerph-19-07374-f001:**
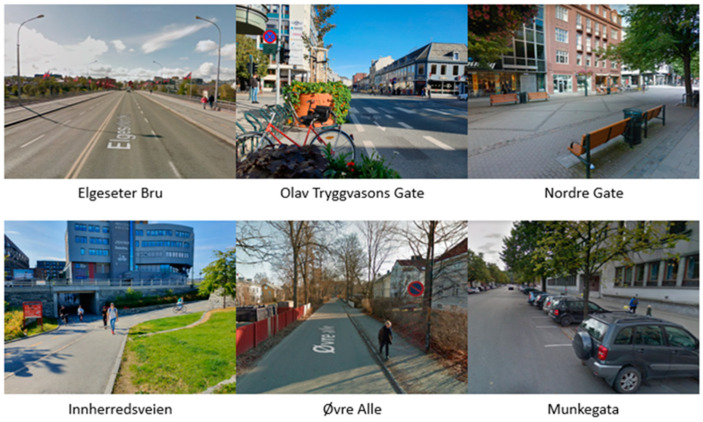
Pictures of the measurement points.

**Figure 2 ijerph-19-07374-f002:**
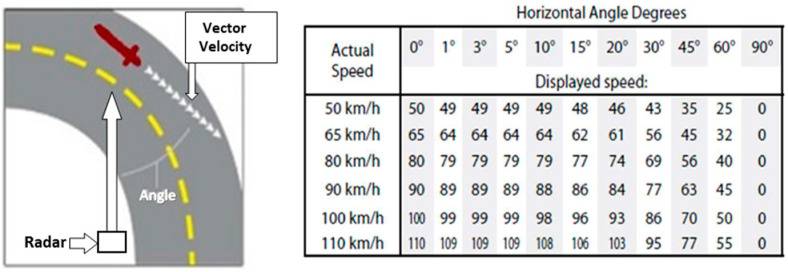
Relationship between antenna target angles and speed.

**Figure 3 ijerph-19-07374-f003:**
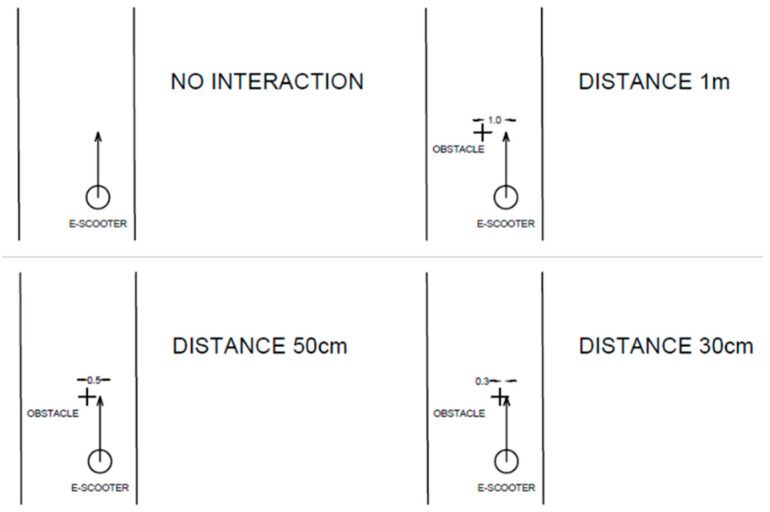
Different types of distances considered.

**Figure 4 ijerph-19-07374-f004:**
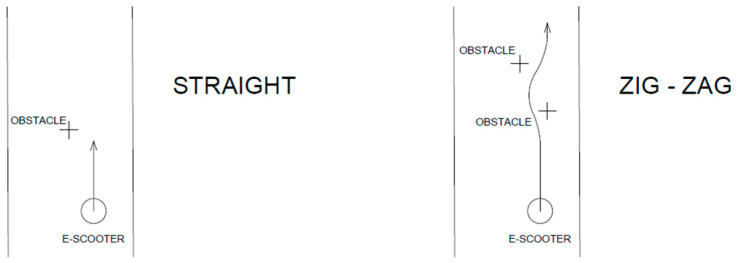
Different types of driver behaviour considered.

**Figure 5 ijerph-19-07374-f005:**
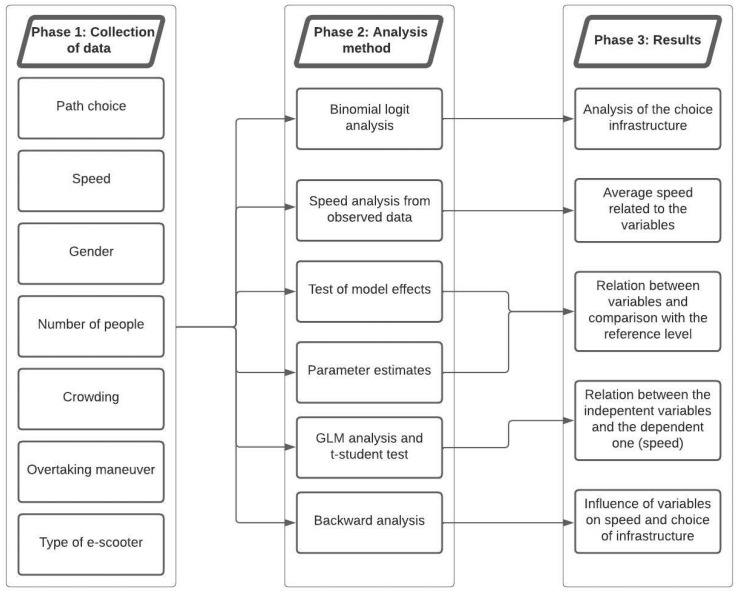
Flow chart of the research phases.

**Figure 6 ijerph-19-07374-f006:**
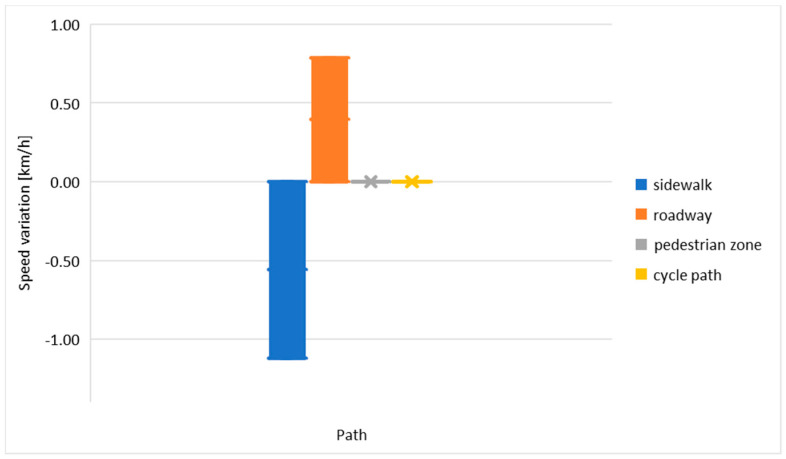
Differences of the absolute value of the speed with reference to path.

**Table 1 ijerph-19-07374-t001:** Tests of model effects.

Tests of Model Effects						
Source	Type I			Type III		
	Wald Chi-Square	df	Sig.	Wald Chi-Square	df	Sig.
(Intercept)	4011.752	1	0.000	253.281	1	0.000
Street	34.623	5	0.000	4.502	4	0.342
Number of people	1.678	1	0.195	1.745	1	0.186
Gender	0.769	1	0.381	1.408	1	0.235
Age	0.808	1	0.369	0.787	1	0.375
Crowding	7.896	3	0.048	5.838	3	0.120
Behaviour	1.198	2	0.549	1.390	2	0.499
Path	2.598	2	0.273	2.598	2	0.273
AddInfo	0.002	1	0.965	0.002	1	0.965

Dependent Variable: Speed; Model: (Intercept); Street; Number of people; Gender; Age; Crowding; Behaviour; Path; AddInfo.

**Table 2 ijerph-19-07374-t002:** Parameter estimates.

Parameter Estimates							
Parameter	B	Std. Error	95% Wald Confidence Interval		Hypothesis Test		
			Lower	Upper	Wald Chi-Square	df	Sig.
(Intercept)	13.96	21.7	9.69	18.22	41.18	1.00	0.00
(Street = OvreAlle)	0.48	22.3	−3.90	4.85	0.05	1.00	0.83
(Street = OlavTrygg)	−0.96	0.82	−2.56	0.63	1.40	1.00	0.24
(Street = Nordre)	−2.92	0.85	−4.59	−1.25	11.69	1.00	0.00
(Street = Munke)	−0.85	12.4	−3.27	1.57	0.47	1.00	0.49
(Street = Inherred)	0.75	0.82	−0.85	2.34	0.84	1.00	0.36
(Street = Elgeseter)	0 ^a^						
(Number of People = 2)	−1.83	13.8	−4.54	0.88	1.75	1.00	0.186
(Number of People = 1)	0 ^a^						
(Gender = M)	0.66	0.56	−0.43	1.75	1.41	1.00	0.24
(Gender = F)	0 ^a^						
(Age = o)	0.58	0.65	−0.70	1.86	0.79	1.00	0.38
(Age = g)	0 ^a^						
(Crowding = distance 50 cm)	0.17	16.1	−2.98	3.33	0.01	1.00	0.91
(Crowding = distance 30 cm)	−1.29	18.9	−5.00	2.42	0.47	1.00	0.49
(Crowding = distance 1 m)	1.90	16.7	−1.38	5.18	1.29	1.00	0.26
(Crowding = a no interaction)	0 ^a^						
(Behaviour = zig-zag)	1.41	13.9	−1.31	4.13	1.03	1.00	0.31
(Behaviour = straight)	2.00	19.2	−1.77	5.76	1.08	1.00	0.30
(Behaviour = reduction speed)	0 ^a^						
(Path = sidewalk)	−1.12	0.79	−2.67	0.43	1.99	1.00	0.16
(Path = roadway)	0.79	18.8	−2.89	4.46	0.18	1.00	0.68
(Path = pedestrian zone)	0 ^a^						
(Path = cycle path)	0 ^a^						
(AddInfo = sharing e-scooter)	−0.03	0.62	−1.25	1.19	0.00	1.00	0.96
(AddInfo = private e-scooter)	0 ^a^						
(Scale)	12.055 ^b^	11.936	9.929	14.637			

Dependent Variable: Speed; Model: (Intercept); Street; Number of People; Gender; Age; Crowding; Behaviour; Path; AddInfo. ^a^. Set to zero because this parameter is redundant. ^b^. Maximum likelihood estimate.

**Table 3 ijerph-19-07374-t003:** Test of model effects without street, Addinfo, and age.

		B	Std. Error	Sig.	
Intercept		13.545	1.983	Value	Diff. Absolute Value
NOP	2	−1.741	1.372	0.205	0.031
	1	0	-		
Gender	M	0.677	0.541	0.211	−0.017
	F	0	-		
Crowding	distance 50 cm	0.478	1.572	0.131	−0.001
	distance 30 cm	−0.869	1.865		
	distance 1 m	2.201	1.652		
	a no interaction	0	-		
Behaviour	zig-zag	1.954	1.303	0.237	0.032
	straight	2.635	1.835		
	reduction of speed	0	-		
Path	sidewalk	−1.909	0.659	0.000	0.000
	roadway	0.723	1.016		
	pedestrian zone	−3.184	0.751		
	cycle path	0	-		

**Table 4 ijerph-19-07374-t004:** Data collected by observer.

Categorical Variable Information				
			N	Percentage
Factor	Street	Overalle	10	4.90
		Olavtrygg	45	22.1
		Nordre	45	22.1
		Munke	15	7.4
		Inherred	45	22.1
		Elgeseter	44	21.6
		Total	204	100.0
	Number of person	2	7	3.4
		1	197	96.6
		Total	204	100.0
	Gender	M	139	68.1
		F	65	31.9
		Total	204	100.0
	Age	18–35	37	18.1
		>35	167	81.9
		Total	204	100.0
	Crowding	distance 50 cm	24	11.8
		distance 30 cm	9	4.4
		distance 1 m	24	11.8
		a no interaction	147	72.1
		Total	204	100.0
	Behaviour	zig-zag	40	19.6
		straight	153	75.0
		reduction speed	11	5.4
		Total	204	100.0
	Path	sidewalk	57	27.9
		roadway	14	6.9
		pedestrian zone	45	22.1
		cycle path	88	43.1
		Total	204	100.0
	Addinfo	sharing e-scooter	159	77.9
		private e-scooter	45	22.1
		Total	204	100.0

**Table 5 ijerph-19-07374-t005:** Average speed and standard deviation.

		N	Min Speed (km/h)	Max Speed (km/h)	Average Speed	Std. Deviation
Dependent Variable	Speed	204	9	27	15.4	3.88

**Table 6 ijerph-19-07374-t006:** Relations between the variables crowding, path, and speed.

Crowding	Average Speed Measured (km/h)			
	Cycle Path	Pedestrian Zone	Sidewalk	Total
No interaction	16.59	12.81	15.00	15.63
Dist. ≤ 50 cm	17.00	12.93	13.00	13.45
Dist. 1 m	16.00	16.00	15.46	15.71
Average	16.59	13.49	14.61	15.26

**Table 7 ijerph-19-07374-t007:** Parameter Estimates for the Path Factor.

Parameter Estimates							
Parameter	B	Std. Error	95% Wald Confidence Interval		Hypothesis Test		
			Lower	Upper	Wald Chi-Square	df	Sig.
(Path = sidewalk)	14.61	0.48	13.67	15.56	924.44	1.00	0.00
(Path = roadway)	17.21	0.97	15.31	19.12	315.04	1.00	0.00
(Path = pedestrian zone)	13.49	0.54	12.43	14.55	621.77	1.00	0.00
(Path = cycle path)	16.59	0.39	15.83	17.35	1839.44	1.00	0.00
(Scale)	13.169 ^a^	1.30	10.85	15.99			

Dependent Variable: Speed; Model: Path; ^a^. Maximum likelihood estimate.

**Table 8 ijerph-19-07374-t008:** *t*-test for the variable Path.

	Average Speed	Sidewalk	Roadway	Ped Zone	Cycle Path
	[km/h]	*t*-Test Value			
Sidewalk	14.61404				
Roadway	17.21429	2.40			
Ped zone	13.48889	1.55	3.35		
Cycle path	16.59091	3.20	0.60	4.66	

**Table 9 ijerph-19-07374-t009:** The test of model effects with statistical significative variables.

		B	Std. Error	Sig.	
Intercept		16.576	0.383	Value	Diff. Absolute Value
Crowding	Distance 50 cm	−0.361	0.838	0.064	0.011
	Distance 30 cm	−2.190	1.281		
	Distance 1 m	1.387	0.851		
	No interaction	0	-		
Path	Sidewalk	−2.029	0.663	0.000	0.000
	Roadway	0.638	1.027		
	Pedestrian zone	−3.081	0.719		
	Cycle path	0	-		

**Table 10 ijerph-19-07374-t010:** Probability to choose the cycle path (Pcp) and sidewalk (Psw).

	Inherredsveien								
	Male	Male	Male	Male	Female	Female	Female	Female	B.A
	Old	Old	Young	Young	Old	Old	Young	Young	
	Shared	Private	Shared	Private	Shared	Private	Shared	Private	
Psw [%]	3.75	4.13	7.71	8.45	2.73	3.00	5.67	6.23	6.67
Pcp [%]	96.25	95.87	92.29	91.55	97.27	97.00	94.33	93.77	93.33
	**Elgeseter**								
	**Male**	**Male**	**Male**	**Male**	**Female**	**Female**	**Female**	**Female**	**B.A**
	**Old**	**Old**	**Young**	**Young**	**Old**	**Old**	**Young**	**Young**	
	**Shared**	**Private**	**Shared**	**Private**	**Shared**	**Private**	**Shared**	**Private**	
Psw [%]	26.89	28.89	44.09	46.55	20.93	22.62	36.20	38.52	38.64
Pcp [%]	73.11	71.11	55.91	53.45	79.07	77.38	63.80	61.48	61.36

**Table 11 ijerph-19-07374-t011:** Probability to choose the cycle path (Pcp) and sidewalk (Psw) for case 2.

	OlavTrygg								
	Male	Male	Male	Male	Female	Female	Female	Female	B.A
	Old	Old	Young	Young	Old	Old	Young	Young	
	Shared	Private	Shared	Private	Shared	Private	Shared	Private	
Psw [%]	41.54	43.97	60.37	62.72	33.83	36.09	52.29	54.76	54.76
Pcp [%]	58.46	56.03	39.63	37.28	66.17	63.91	47.71	45.24	45.24

## Data Availability

The data presented in this study are available on request from the corresponding author.

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
