# Peer review of "New Micromobility Means of Transport: An Analysis of E-Scooter Users’ Behaviour in Trondheim"

_ijerph, 2022, doi:10.3390/ijerph19127374_

Round 1

Reviewer 1 Report

Dear Editor,

The manuscript entitled "New micromobility means of transport: an analysis of e-scooter 2 users’ behaviour and speed in Trondheim” is an interesting study and I really enjoyed reading the paper. However, in my opinion, the manuscript has some shortcomings in the text. According to the mentioned items, I recommend Major revision for the manuscript.

1.      Please clarify the research gap as well as the innovation of this research at the end of Introduction section.

2.      Figures 5 and 6 have not been cited in the text.

3.      Table 3 has not been cited in the text.

4.      The authors should use more references in 2021 and 2022. There is no paper in these years. Unfortunately, the paper hasn’t had strong literature review. They can study and address the followings papers:

https://doi.org/10.1016/j.ijtst.2022.02.001

https://doi.org/10.1177%2F03611981221083306

https://doi.org/10.3390/su14052564

https://doi.org/10.1016/j.trd.2022.103229

https://doi.org/10.1016/j.aap.2022.106685

https://doi.org/10.1016/j.jtrangeo.2021.103016

https://doi.org/10.1016/j.trd.2021.102790

https://doi.org/10.1016/j.aap.2020.105954

https://doi.org/10.1016/j.tra.2021.09.012

https://doi.org/10.1016/j.trd.2021.102844

5.      Try to polish the writing and English of the manuscript. I found several errors.

6.      The authors should provide quantitative results in both the abstract as well as conclusion section. For example: the average speed of e-scooter of present study

7.      The author should describe the detail of study via comprehensive flow chart in material and method section.

8.      It is compulsory to provide results of the speed differences between males and females or other characteristic parameters via T-test or ANOVA according to following paper:

https://doi.org/10.1155/2022/7768160

9.      The conclusion section should represent the results on a case-by-case basis by bullets.

10.  It is better authors compare their results with previous studies of e-scooter in order to present state of art and it is also suggested the authors compare the result of present studies with pedestrians’ speed in similar facilities. For example: the pedestrian speed with e-scooter speed in side walk according to the followings papers:

https://doi.org/10.1088/1757-899X/245/4/042014

https://doi.org/10.1007/s11709-021-0785-x

-          Estimated Total Stadium Emergency Evacuation’s Time Based on Existing Population Analysis, Computational Research Progress in Applied Science & Engineering (CRPASE)

-          Recommended walking speeds for timing of pedestrian clearance intervals based on characteristics of the pedestrian population. Transportation research record

-          Field studies of pedestrian walking speed and start-up time. Transportation research record,

-          Evaluation of pedestrians speed with investigation of un-marked crossing.

-          Walking Speeds of Ederly Pedestrians at Crosswalks. Transportation Research Record

-          Pedestrian flow characteristics in Hong Kong. Transportation Research Record

-          Analysis of the effect of vehicles conflict on pedestrian's crossing speed in signalized and un-signalized intersection. Advances in Environmental Biology.

Author Response

The manuscript entitled "New micromobility means of transport: an analysis of e-scooter 2 users’ behaviour and speed in Trondheim” is an interesting study and I really enjoyed reading the paper. However, in my opinion, the manuscript has some shortcomings in the text. According to the mentioned items, I recommend Major revision for the manuscript.

  1. Please clarify the research gap as well as the innovation of this research at the end of Introduction section.

In the first part of introduction the authors have described the state of art about the current situation of scooters highlighting the related criticalities also in relation to the references. Then at the end of this section the authors have defined the searched results of the field analysis. In particular the practical aim of the study has been presented in the last rows: “Moving on the empiric point of view, this research aims to contribute to the literature through data obtained from field analyses of speed and behaviour, in terms of choice and interaction with the other weak users of the road. Moreover, the identification of the factors choice of the path may help local politicians to develop policies and rules based on concrete evidence and mobility planners to develop actions for better integration of e-scooters in cities.” (From row 117 to 122). The authors with these words identified the aim to identify how (in terms of speed) and which (in terms of path choice) are the used infrastructures. In this way the authors wanted to drive the local policies through the safe development of e-scooter.

  1. Figures 5 and 6 have not been cited in the text.

The authors provided to change like you kindly noted (Figure 5 has been replaced with the flow chart of the research in row 302).

  1. Table 3 has not been cited in the text.

The authors provide to insert missing table.

  1. The authors should use more references in 2021 and 2022. There is no paper in these years. Unfortunately, the paper hasn’t had strong literature review. They can study and address the followings papers:
  • https://doi.org/10.1016/j.ijtst.2022.02.00: Agent-based models for simulating e-scooter sharing services: A review and a qualitative assessment
  • https://doi.org/10.1177%2F03611981221083306: Shared E-Scooter Trajectory Analysis During the COVID-19 Pandemic in Austin, Texas
  • https://doi.org/10.3390/su14052564: Open AccessArticle
  • Predicting Demand for Shared E-Scooter Using Community Structure and Deep Learning Method
  • https://doi.org/10.1016/j.trd.2022.103229: : Comprehensive comparison of e-scooter sharing mobility: Evidence from 30 European cities
  • https://doi.org/10.1016/j.aap.2022.106685: Secondary task engagement, risk-taking, and safety-related equipment use in German bicycle and e-scooter riders – An observation
  • https://doi.org/10.1016/j.jtrangeo.2021.103016: Spatial analysis of shared e-scooter trips
  • https://doi.org/10.1016/j.trd.2021.102790: Analysis of attitudes and engagement of shared e-scooter users
  • https://doi.org/10.1016/j.aap.2020.105954: E-Scooter safety: The riding risk analysis based on mobile sensing data
  • https://doi.org/10.1016/j.tra.2021.09.012: Impact of e-scooter sharing on bike sharing in Chicago
  • https://doi.org/10.1016/j.trd.2021.102844: The relationship between E-scooter travels and daily leisure activities in Austin, Texas

The authors thank you for these references. The following papers has been added in order to refer the research to the current situation:

  • “https://doi.org/10.1177%2F03611981221083306: Shared E-Scooter Trajectory Analysis During the COVID-19 Pandemic in Austin, Texas” with reference 30;
  • “ https://doi.org/10.1016/j.trd.2022.103229: Comprehensive comparison of e-scooter sharing mobility: Evidence from 30 European cities” with reference 14;
  • https://doi.org/10.1016/j.trd.2021.102790: Analysis of attitudes and engagement of shared e-scooter users” with reference 22;
  • https://doi.org/10.1016/j.aap.2020.105954: E-Scooter safety: The riding risk analysis based on mobile sensing data” with reference 27;
  • https://doi.org/10.1016/j.tra.2021.09.012: Impact of e-scooter sharing on bike sharing in Chicago” with reference 9.
  1. Try to polish the writing and English of the manuscript. I found several errors.

The authors thank you for the suggestion. The all manuscript have gone through a new editing by a technical English native Agency.

  1. The authors should provide quantitative results in both the abstract as well as conclusion section. For example: the average speed of e-scooter of present study.

The suggested modification has been executed. Quantitative results have been added in order to clarify the exposition. In the abstract the authors have indicated the average speed (row 21) and the percentage value to choose the cycle path (row 26). In addition, the conclusion section has been improved with the insert of the speed and behaviour results (from row 570 to 583) in according to your kindly reviews.

  1. The author should describe the detail of study via comprehensive flow chart in material and method section.

The authors thank for the suggestion. The flow chart of research has been added in order to clarify the steps of the study.

  1. It is compulsory to provide results of the speed differences between males and females or other characteristic parameters via T-test or ANOVA according to following paper:

https://doi.org/10.1155/2022/7768160: Pedestrians Crossing and Walking Speeds Analysis in Urban Areas under the Influence of Rain and Personality Characteristics

The authors provided to add the suggested reference. The two papers have been compared for the T-test analysis (row 206).

  1. The conclusion section should represent the results on a case-by-case basis by bullets.

The suggested modification has been executed.

  1. It is better authors compare their results with previous studies of e-scooter in order to present state of art and it is also suggested the authors compare the result of present studies with pedestrians’ speed in similar facilities. For example: the pedestrian speed with e-scooter speed in side walk according to the followings papers:
  • https://doi.org/10.1088/1757-899X/245/4/042014: Unmarked Crosswalks in the Signalized and Un-Signalized Intersections (Case Study: Rasht city)
  • https://doi.org/10.1007/s11709-021-0785-x: Presentation of regression analysis, GP and GMDH models to predict the pedestrian density in various urban facilities
  • Field studies of pedestrian walking speed and start-up time. Transportation research record,
  • Evaluation of pedestrians speed with investigation of un-marked crossing.
  • Walking Speeds of Ederly Pedestrians at Crosswalks. Transportation Research Record
  • Pedestrian flow characteristics in Hong Kong. Transportation Research Record
  • Analysis of the effect of vehicles conflict on pedestrian's crossing speed in signalized and un-signalized intersection. Advances in Environmental Biology.

The authors provided to add the suggested reference:

  • https://doi.org/10.1088/1757-899X/245/4/042014: Unmarked Crosswalks in the Signalized and Un-Signalized Intersections (Case Study: Rasht city) with reference 20.

Reviewer 2 Report

The manuscript is well structured and well argued. However, several rectifications and modifications are required to ensure that its quality stands up to this reputed journal.  

  1. The authors have presented the speed and behaviour analysis of e- scooter riders and their interactions with other road users to understand the management of transport as per the government policies.
  2. The English language must be improved. There are several grammatical errors as one goes through the manuscript that requires rectification. Most of the sentences convey no proper meaning and could be off-putting to the readers and practitioners.
  3. The first section introduces a basic outlook on use of micromobility means of transport in various countries and a brief literature review of surveys on users conducted in different countries is presented.
  4. Authors have to clearly mention the details of Generalized Linear Model (GLM) and SPSS program preferably with flow chart in order to help the readers to understand clearly. Why authors have specifically chosen GLM for the analysis and Binomial Logit Model for the pavement choice?
  5. To keep things fair, a brief discussion of the demerits with the proposed analysis should be provided.

  1. The manuscript has the potential to be improved and requires minor rectifications. With that being said, I wish the authors all the best on their endeavor to improve the quality of the manuscript.

The paper is not acceptable in the current form but can be after minor modifications as suggested above.

Author Response

The manuscript is well structured and well argued. However, several rectifications and modifications are required to ensure that its quality stands up to this reputed journal. The authors have presented the speed and behaviour analysis of e- scooter riders and their interactions with other road users to understand the management of transport as per the government policies.

  1. The English language must be improved. There are several grammatical errors as one goes through the manuscript that requires rectification. Most of the sentences convey no proper meaning and could be off-putting to the readers and practitioners.

The authors thank you for the suggestion. The document was completely revised, any grammatical errors were corrected, and sentences rectified to convey proper meaning.

  1. The first section introduces a basic outlook on use of micromobility means of transport in various countries and a brief literature review of surveys on users conducted in different countries is presented.
  2. Authors have to clearly mention the details of Generalized Linear Model (GLM) and SPSS program preferably with flow chart in order to help the readers to understand clearly. Why authors have specifically chosen GLM for the analysis and Binomial Logit Model for the pavement choice?

At the end of paragraph 2, a flow chart with the phases of the study and the methodology used was inserted. The GLM is a statistical method to analyse data and find correlations between the different elements. The GLM expands the general linear model so that the dependent variable is linearly related to the other factors and covariates by a specified link function (relationship between the linear predictor and the mean of the distribution function). Linear regression predicts the expected value of a given unknown quantity (the response variable) as a linear combination of a series of observed values. This implies that a constant change of a considered factor leads to a constant change in the response variable. The GLM model is appropriate when the response variable has a normal distribution, to describe a random variable with real values that tend to concentrate around a single average value. For this reason, the GLM was used in the analysis to evaluate the relationship between the speed as dependent variable and the independent variables. A modal analysis was used to understand which type of infrastructure is preferred by the e-scooter users. In the SPSS program, to make this modal analysis, the Binomial Logit model was used considering Path like the dependent variable, with only Sidewalk and Cycle Path inside. The following parameters were chosen that could influence the choice: Street, Gender, Age and AddInfo. In the program the option Binary logistic was chosen as scale response, where the distribution is the Binomial and the link function is the Logit one. The Binomial Logit Model calculates the probability that an observation falls on one or the other category of dependent variable (in our case it is the probability of choosing the cycle path or the sidewalk) based on one or more independent variables that can be continuous or categorical (such as Street, Gender, Age and AddInfo).

  1. To keep things fair, a brief discussion of the demerits with the proposed analysis should be provided.

Demerits of the analysis and possible ideas for future research were written in the Conclusions. The number of e-scooters analysed, divided between shared and non-shared, could be increased to reach a greater accuracy in the analysis. In addition, it could be very interesting to simulate e-scooters in city-centre traffic through a microsimulation, especially in situations of extreme crowding. This data could be useful to better study and fully understand the problem of the distance maintained by e-scooter riders especially during overtaking. Additional data could also lead to the best solutions to improve the coexistence of e-scooters with pedestrians and cyclists and help identify right places to install parking lots.

The manuscript has the potential to be improved and requires minor rectifications. With that being said, I wish the authors all the best on their endeavor to improve the quality of the manuscript.

The paper is not acceptable in the current form but can be after minor modifications as suggested above.

Reviewer 3 Report

This is a potentially important paper as these micro-vehicles are quickly springing up around the world and do offer an alternative to encourage mobility shifts and safety improvements. However, the paper lists only behaviour and speed. In its current form, it is too long and complex and needs a substantial edit to focus on the key behaviour and speed topics in the paper’s title. Other comments are listed below.

·       The abstract is around 420 words and is far too long (200 words maximum)

·       While the title lists behaviour and speed, many other aspects seem to have been included in the analysis.

·       No hypothesis or hypotheses are noted to identify the theme of this paper.

·       Like the abstract, the paper would benefit by a substantial reduction in size. For instance, there is much discussion about the statistical approach, much of which should be put into an appendix. 

·       The use of more paragraphs to break up the text into relevant sections will improve the paper’s readability.

·       More details would be useful regarding the observation periods, what criteria were used in selecting the trial areas, and how long were the observation periods. The influence of the presence of the observer and any attempt to hide should also be mentioned.

·       Rider age validity should be mentioned and >35years is not “elderly”

·       The statistic tests used were excellent although in the result section there were no statics given for the differences in the interaction analyses, hence the reader is left wondering how significant these finding were?

·       The writing is generally good, although the results should be written in past-tense.

·       Comparison with previous findings in France, Sweden, and elsewhere is good but without knowing more about the trial similarities, it is difficult to appreciate this.  

·       The discussion section is too long as is the conclusion section. I’d like to see these combined and shortened for the discussion only. Specifying strengths and limitations with the research and further work to be addressed would be helpful.

·       In regard of my last point, the behaviour is assumed to be safer than other modes of transport. I’m not sure that this is so, especially given your findings for crowding and path use. This is surely something worthy of further research.  

Conclusions should be brief (no more than 100-200 words), with a number of relevant dot points in respect of the aims and hypotheses.

Author Response

This is a potentially important paper as these micro-vehicles are quickly springing up around the world and do offer an alternative to encourage mobility shifts and safety improvements. However, the paper lists only behaviour and speed. In its current form, it is too long and complex and needs a substantial edit to focus on the key behaviour and speed topics in the paper’s title. Other comments are listed below.

  1. The abstract is around 420 words and is far too long (200 words maximum)

After revision, the abstract is shorter than the previous article.

  1. While the title lists behaviour and speed, many other aspects seem to have been included in the analysis.

The title of the article has been changed by deleting the word "speed". The choice of authors is due to the fact that the "behaviour" of users driving e-scooters can include all the other elements analysed.

  1. No hypothesis or hypotheses are noted to identify the theme of this paper.

The initial hypothesis of the study was to understand if the behaviour of e-scooters drivers could be considered similar to that of cyclists in order to be able to use the same policies of bicycles also for e-scooters within the cities.

  1. Like the abstract, the paper would benefit by a substantial reduction in size. For instance, there is much discussion about the statistical approach, much of which should be put into an appendix. 

The authors have completely revised the paper by eliminating redundant parts. According to the suggestion of other reviewers, the description of the statistical analysis was not included in an appendix as it is crucial to better understand the results obtained and shown in the next paragraph.

  1. The use of more paragraphs to break up the text into relevant sections will improve the paper’s readability.

The authors are very grateful for the suggestion provided. They divided the text into shorter sections according to their content and tried to summarize and outline the paragraphs in order to make the analysis clearer without adding new ones.

  1. More details would be useful regarding the observation periods, what criteria were used in selecting the trial areas, and how long were the observation periods. The influence of the presence of the observer and any attempt to hide should also be mentioned.

In paragraph 2 - "Materials and Methods", the authors included more information about the period of observation, the influence of the observer’s presence and other useful aspects to understand the way data were collected.  To understand the behaviour of e-scooter drivers, six sections of different roads in the city centre of Trondheim were analysed. The downtown streets were selected for analysis since here it was more likely to find e-scooters in crowded situations interactiong with other road users. Pedestrians and cyclists were the most interested groups in the analysis as they used the same infrastructure as the e-scooters. Just for these features, streets in the city centre were chosen or roads connecting the suburbs and the centre. A traffic radar was used to detect the speed of e-scooters. The presence of a human observer also helped to grasp some important aspects of each situation/context. In order not to affect the behaviour of drivers, the radar instrument was installed in a hidden point, adjacent to the section to be analysed so as to measure the real speed of e-scooters.  Otherwise, the data obtained would not have been realistic enough.

  1. Rider age validity should be mentioned and >35years is not “elderly”

Authors have changed both aspects within the article.

  1. The statistic tests used were excellent although in the result section there were no statics given for the differences in the interaction analyses, hence the reader is left wondering how significant these finding was?

The results obtained from the statistical analysis were reported and discussed in chapter 3 - "Results". They were divided into sub-paragraphs according to the type of analysis carried out. In particular, the results obtained can be consulted in Tables 5,6,7,8,9,10 and 11.

  1. The writing is generally good, although the results should be written in past-tense.

All results obtained are reported in past tense now.

  1. Comparison with previous findings in France, Sweden, and elsewhere is good but without knowing more about the trial similarities, it is difficult to appreciate this.

Authors added in paragraph 3.1 more details on previous findings in other countries to emphasize the similarities and differences between the studies.

  1. The discussion section is too long as is the conclusion section. I’d like to see these combined and shortened for the discussion only. Specifying strengths and limitations with the research and further work to be addressed would be helpful.

The "discussion" section has been shortened as well as the conclusions specifying strengths and limits of the study. For example, the number of analysed e-scooters, divided between shared and non-shared, could be increased in order to reach a greater precision in the analysis. In addition, it could be very interesting to simulate e-scooters in city-centre traffic through a microsimulation, especially in situations of extreme crowding. This data could be useful to deepen the problem of the distance maintained by e-scooter riders, especially during overtaking. But they could also be useful to decide what solutions to take to improve the coexistence of e-scooters with pedestrians and cyclists, as well as the best places to install parking lots.

  1. In regard of my last point, the behaviour is assumed to be safer than other modes of transport. I’m not sure that this is so, especially given your findings for crowding and path use. This is surely something worthy of further research.

Authors would like to thank the reviewer for his helpful suggestion. The analysis will certainly be considered for future research.  

  1. Conclusions should be brief (no more than 100-200 words), with a number of relevant dot points in respect of the aims and hypotheses.

The conclusions were shortened, and a bulleted list highlights the salient results achieved thanks to the analysis carried out.

Reviewer 4 Report

This study investigated the speed of e-scooter users and their interactions with other road users based on the real-world data, and proposed some policy implications for policy-makers to better integrate the e-scooters into cities. Overall, the manuscript is in good shape and requires minor revisions.

1. E-scooter is also an important transportation mode in China. However, there is a lack of traffic behavior analysis of e-scooters (e-bikers) in China. Some important literature related to the development of e-bikes should be added in literature review:

(1) Understanding electric bike riders’ intention to violate traffic rules and accident proneness in China.

(2) Analysis of crossing behavior and violations of electric bikers at signalized intersections.

(3) Understanding on-road practices of electric bike riders: An observational study in a developed city of China.

(4) The red-light running behavior of electric bike riders and cyclists at urban intersections in China: An observational study.

2. The practical valuable of this paper is not very clear. This study proposed some policy implications, but the findings/recommendations are obvious and did not put forward any new suggestions to the current state of practice. The authors need to further clarify the practical valuable of their work.

3. Minor language typos should be corrected; lack of commas; long sentences should be rewritten (it is hard to follow). Presentation of the results can be less dense; the authors are encouraged to synthesize information.

4. The resolution of all figures is poor. In addition, car graphics need to be replaced by e-scooter graphics in Figure 2.

Author Response

This study investigated the speed of e-scooter users and their interactions with other road users based on the real-world data and proposed some policy implications for policy-makers to better integrate the e-scooters into cities. Overall, the manuscript is in good shape and requires minor revisions.

  1. E-scooter is also an important transportation mode in China. However, there is a lack of traffic behavior analysis of e-scooters (e-bikers) in China. Some important literature related to the development of e-bikes should be added in literature review:
    • Understanding electric bike riders’ intention to violate traffic rules and accident proneness in China.
    • Analysis of crossing behavior and violations of electric bikers at signalized intersections.
    • Understanding on-road practices of electric bike riders: An observational study in a developed city of China.
    • The red-light running behavior of electric bike riders and cyclists at urban intersections in China: An observational study.

The authors provided to add the suggested reference, increasing the information about the behaviour of the different type of the users on the street.

  • “The red-light running behaviour of electric bike riders and cyclists at urban intersections in China: An observational study” with reference 19 (row 332).
  1. The practical valuable of this paper is not very clear. This study proposed some policy implications, but the findings/recommendations are obvious and did not put forward any new suggestions to the current state of practice. The authors need to further clarify the practical valuable of their work.

The authors thank for the revision. According to the data provided by the research, in the final part of the introduction (from 573th to 582th row) some suggestions are added: “Additional data on the relationship between the behaviour of e-scooters and interactions with other users and cyclists could be collected with a survey through the app used to rent e-scooters and installing cameras. Information about the busiest hours of the day, the average speed and the length of each trip may also be collected. In addition, it could be very interesting to simulate e-scooters in city-centre traffic through a microsimulation, especially in situations of extreme crowding. This data could be useful to deepen the problem of the distance maintained by e-scooter riders especially during overtaking. But they could also be useful to decide what solutions to take to improve the coexistence of e-scooters with pedestrians and cyclists, as well as the best places to install parking lots.”. The authors in this section would suggest how the co-operation between the local authorities and rent companies could be given more information about the e-scooter development in the cities. In fact, with the practical suggestions provided in this section, it will be possible to calibrate both the offer, the infrastructure and the availability of parking spots.

  1. Minor language typos should be corrected; lack of commas; long sentences should be rewritten (it is hard to follow). Presentation of the results can be less dense; the authors are encouraged to synthesize information.

The authors thank for the suggestions. The conclusion section has been summarized, using also a bullet scheme, in order to focus on the significant results obtained. In addition to this, all the paper has been revised with a view to increase the fluidity of the sentences especially in the “discussion” section.

  1. The resolution of all figures is poor. In addition, car graphics need to be replaced by e-scooter graphics in Figure 2.

The authors provide to modify the Figure 2 (row 161). Also the resolution of all images has been improved.

Round 2

Reviewer 1 Report

Dear editor
All grammatical errors and comments have been revised by the authors.
Regards,